# Emotional Regulation of Displaced Aggression in Provocative Situations among Junior High School Students

**DOI:** 10.3390/bs14060500

**Published:** 2024-06-14

**Authors:** Shuang Lin, Gonglu Cheng, Shinan Sun, Mengmeng Feng, Xuejun Bai

**Affiliations:** 1Faculty of Psychology, Tianjin Normal University, Tianjin 300387, China; lins000@126.com (S.L.); chgonglu@126.com (G.C.); psyssn@126.com (S.S.); psyfengmm@126.com (M.F.); 2Key Research Base of Humanities and Social Sciences of the Ministry of Education, Academy of Psychology and Behavior, Tianjin Normal University, Tianjin 300387, China

**Keywords:** provocative situation, emotional regulation, displaced aggression, junior high school students

## Abstract

This study investigated the emotion regulation effect of displaced aggression among junior high school students after provocation through two experiments. Experiment 1 examined the effect of displaced aggression on the negative and positive emotions of junior high school students after low- and high-level provocation. The results showed that only after high-level provocation did individuals experience a significant decrease in negative emotions and a significant increase in positive emotions after engaging in displaced aggression. Experiment 2 explored the effect of aggressive intensity on negative and positive emotions after provocation. The results indicated that, in terms of changes in positive emotions, low-intensity aggression showed a significant increase in positive emotions after aggression. Regarding changes in negative emotions, both low-intensity aggression and high-intensity aggression resulted in significant decreases in negative emotions after aggression. In conclusion, this research showed that, in highly provocative situations, displaced aggression among junior school students, especially low-intensity displaced aggression, could increase positive emotions and decrease negative emotions. These results support the emotional regulation theory of aggression. However, considering that displaced aggression violates social norms, efforts should be made to avoid individuals regulating their emotions through displaced aggression, instead guiding them toward using more appropriate methods for emotional regulation in future research and practical applications.

## 1. Introduction

Aggression refers to any behavior directed toward another individual that is carried out with the proximate (immediate) intent to cause harm [1]. Depending on the reasons and intentions behind aggression, it can be classified into reactive aggression and proactive aggression. Reactive aggression refers to uncontrolled aggression that individuals exhibit in response to stimuli or provocation. This includes direct retaliatory reactive aggression, where individuals retaliate directly against the provocateur, and displaced aggression, where individuals, instead of retaliating directly against the provocateur, target innocent individuals. Proactive aggression refers to planned, intentional, and voluntary aggression [2]. Displaced aggression, as a specific type of reactive aggression, differs from direct retaliatory reactive aggression in that it involves individuals refraining from directly retaliating against the provocateur after experiencing frustration or provocation and instead targeting innocent individuals. Displaced aggression, involving provocation from provocateurs and retaliation against innocent individuals [3], is prevalent in various settings such as workplaces [4], schools [5], and families [6]. Overall, displaced aggression shows a trend of initially increasing and then decreasing, peaking during adolescence [7]. The period of junior high school marks the transition from childhood to adolescence, where individuals exhibit an imbalance in their cool and hot systems, with the hot system developing faster than the cool system [8], leading to the development of impulsive and aggressive psychological characteristics [9]. A study involving 1879 junior high school students found that although only 11.5% of students admitted to having attacked other classmates, 46.4% reported experiencing unprovoked attacks by classmates at least 1–2 times in the past [10].

Generally speaking, aggression, especially displaced aggression, is considered maladaptive and antisocial behavior with major social harm [11]. However, aggression also serves certain adaptive functions [12]. These adaptive functions play a role in the initiation and perpetuation of aggression. Catharsis theory and related research suggest that aggression could help individuals release their negative emotions and improve their positive emotions [13]. In other words, aggression could be driven by the motive to eliminate or reduce aversive stimuli, and the release of negative emotions, to some extent, reflects the adaptive function of aggression. However, the emotional regulation theory suggests that aggression can not only alleviate negative emotions but also enhance positive emotions [14]. Some studies on the relationship between sadism and aggression have strongly supported this point [15]. Sadistic tendencies involve experiencing positive emotions during the process of harming others, emphasizing the anticipation of positive emotions derived from aggression rather than the positive emotions themselves.

From the existing perspectives, the emotion-regulating effects of different types of aggression vary. Firstly, in terms of positive emotions, previous research has found that individuals experience a significant increase in positive emotions after engaging in reactive aggression and displaced aggression. Specifically, some studies have indicated that individuals are more likely to experience positive emotions after engaging in reactive aggression [16]. Neuroscientific research also supports this view, as it has been found that reward systems, such as the ventral striatum, are significantly activated when engaging in reactive aggression [17]. Furthermore, some studies have supported the emotion-regulating function of displaced aggression; for example, a study found that males exhibited more intense aggression towards other females after provocation by a female and sought to experience a sense of revenge pleasure [18]. Neuroscientific research has also indicated a significant positive correlation between the orbitofrontal cortex (OFC), a brain region associated with aggression motivation and reward processing, and the revenge plans of displaced aggression [19]. Secondly, inconsistent results regarding negative emotions have been obtained. Some studies have suggested that reactive aggression could reduce negative emotions [20]. However, displaced aggression may not decrease negative emotions. Specifically, displaced aggression, as a special form of reactive aggression produced by individuals in response to provocation, is primarily aimed at venting the negative emotions caused by provocation [21]. Therefore, displaced aggression may have the potential to reduce negative emotions. However, recent viewpoints suggest that venting may exacerbate negative emotions rather than reducing them [22]. Furthermore, displaced aggression involves attacking innocent individuals, which seriously violates social norms and may place moral pressure on individuals.

Considering that displaced aggression is triggered by provocation, individuals may have different initial emotional levels and aggression intensities in different provocative situations, leading to varying effects on emotion regulation [23]. Specifically, there are individual differences in the regulating effects of positive emotions in low provocative situations, which may be related to certain antisocial personality traits [24]. For example, individuals with a high tendency for cruelty may view attacking innocent individuals as a form of entertainment and derive positive emotions from displaced aggression [25]. Furthermore, previous research found that individuals had lower levels of negative emotion arousal in low provocative situations, and the role of emotional regulation through displaced aggression was less significant in this context [26]. On the contrary, in highly provocative situations, individuals might turn to displaced aggression as a means of regulating emotions due to experiencing higher levels of negative emotions and lower levels of positive emotions after provocation [27]. Specifically, this research used social exclusion as a provocative situation and voodoo dolls as substitutes for individuals who were being socially excluded, examining changes in participants’ emotions before and after aggression. The results showed that, after provocation, the participants experienced high levels of negative emotions, and positive emotions increased after attacking the voodoo doll. However, this study did not specifically investigate changes in positive and negative emotions before and after aggression, and voodoo dolls do not truly represent innocent individuals. Therefore, researchers need to further examine the regulatory effects of displaced aggression in provocative situations through experimental methods. Based on emotion regulation theory, this study proposed hypothesis H1: junior high school students will experience regulatory effects after engaging in displaced aggression after high-level provocation, specifically manifesting as a decrease in negative emotions and an increase in positive emotions.

Furthermore, provocative situations can increase the intensity of individual aggression, and a study also found that the higher the level of provocation, the greater the intensity of displaced aggression [28]. Meanwhile, some studies suggest that different levels of aggression have different effects on emotional regulation [29]. The opponent-process theory suggests that initial aggression generates negative emotions, and in order to counteract these negative emotions, individuals have to spontaneously generate positive emotions [30]. However, the initially generated positive emotions are weak, and with an increase in the number of aggressive acts, negative emotions decrease while positive emotions increase. In the process of confrontation, positive emotions would dominate and become the emotions associated with aggression. During the process of engaging in displaced aggression, since the targets of aggression are innocent individuals, it is not easy for individuals to experience positive emotions. In fact, they may feel guilt and other negative emotions, and the greater the intensity of aggression, the stronger the negative emotions that individuals may experience. Attacking innocent individuals in displaced aggression does not align with the tit-for-tat matching rule, and society has less tolerance for such behavior. Therefore, the higher the intensity of aggression toward innocent individuals, the less likely it is to produce pleasure effects. Based on emotion regulation theory and the opponent-process theory, this study proposed hypothesis H2: after provocation, low-intensity aggression towards innocent individuals by junior high school students will have a stronger regulatory effect compared to high-intensity aggression, specifically manifested as a decrease in negative emotions and an increase in positive emotions.

Overall, this study included two experiments to examine the regulatory effects of displaced aggression after provocation. Experiment 1 investigated the effect of displaced aggression by junior high school students on emotions after provocation. Specifically, a two-stage competitive reaction time task was used [31], with high- and low-level provocation situations in the first stage (i.e., passive stage) and an assessment of displaced aggression among junior high school students in the second stage (i.e., active stage). Changes in negative and positive emotions were measured before and after aggression to test the regulatory effects of displaced aggression on emotions. Then, building upon Experiment 1, Experiment 2 manipulated the intensity of aggression after high-level provocation to examine the effect of aggression intensity on emotions.

## 2. Experiment 1: The Effect of Displaced Aggression on Emotions after Provocation among Junior High School Students

Experiment 1 used a two-stage competitive reaction time task, manipulating provocative situations in the passive stage, and measuring positive and negative emotions before and after the active stage. The purpose of Experiment 1 was to explore the regulatory effects of the displaced aggression of junior high school students after experiencing different provocative situations and test hypothesis H1.

### 2.1. Methods

#### 2.1.1. Participants

We estimated the required sample size using G*Power 3.1. Experiment 1 employed a 2 × 2 within-subject design, with *f* = 0.25, *α* = 0.05, and a statistical power of 0.8; the estimated sample size was 24. Based on the principle of convenience sampling, thirty-six junior high school students were recruited from a junior high school in Zhuzhou City, Hunan Province, China. According to the experimental requirements, three students with single-key reactions, two students who did not distinguish provocateurs and innocent targets, and those who doubted the authenticity of opponents were excluded. Finally, thirty-one participants remained in the formal sample for analysis. The formal sample consisted of 17 males, aged 12 to 14, with an average age of 12.71 years (*SD* = 0.69), and 14 females, aged 12 to 14, with an average age of 12.64 years (*SD* = 0.63). All of the participants had normal or corrected vision, no color blindness, and no history of mental illness or surgical trauma. The experiment took place from April to May 2023.

#### 2.1.2. Design

A 2 (provocation: low provocation and high provocation) × 2 (measurement phase: prior to aggression and post-aggression) within-subject design was employed. In the case of the low provocation condition, the punisher’s selection of high punishment occurred 20% of the time, and the participants’ acceptance of punishment happened 20% of the time. In the high provocation condition, the punisher’s selection of high punishment occurred 80% of the time, and the participants’ acceptance of punishment happened 80% of the time. The dependent variables were positive emotions and negative emotions, as scored on the PANAS positive emotion sub-scale and negative emotion sub-scale.

#### 2.1.3. Materials

The modified competitive reaction time paradigm was used to investigate individuals’ displaced aggression [31]. This paradigm consisted of two stages: a passive stage and an active stage. In the passive stage, the participant would be provoked by an opponent but could not retaliate. Specifically, in the passive stage, the participant was required to compete against opponent A within a specified time frame. Participants saw the portrait of opponent A on the first screen in the passive stage and were aware that they would compete with opponent A later in this stage. When a white dot appeared on the screen, both players had to press the space bar as quickly as possible, and the faster reactor would win. However, in the passive stage, the participant would receive punishment from opponent A if they lost, but even if they won, they could not punish opponent A. Provocation by the opponent at the passive stage had two levels: high provocation and low provocation, and the order of presenting high and low provocation levels was counterbalanced among the participants. Under high provocation conditions, the opponent would choose high punishment in 80% of trials and low punishment in 20%. Under low provocation conditions, the opponent would choose low punishment in 80% of trials and high punishment in 20% of trials [32].

Before transitioning to the active stage (second stage), we explicitly informed the participants that they would not continue competing with opponent A, as shown in Figure 1. In the active stage, we ensured that the participants understood they would be facing opponent B. Compared to the passive stage, in the active stage, participants would not receive any punishment after winning the match and could punish the opponent according to a pre-selected intensity, but they could not punish the opponent if they lost the match, as shown in Figure 2. In the post-experimental interview, two questions were included: (1) whether the participants thought they faced different opponents in the passive and active stages, and (2) the reasons for the participants’ choice of punishment level in the active stage. The interview results indicated that all participants believed that the participants in the two stages were different, and none of the participants refrained from attacking the innocent opponent in the active stage because the opponent was an innocent party (not responsible for provoking the participant in the passive stage).

After consulting the relevant literature using the competitive reaction time paradigm [33] and the threshold of physiological harm from white noise [34], this study utilized white noise as a form of punishment. Twelve different intensity levels of white noise patterns were created using “Cool Edit 2.1” software, ranging from 55 dB to 110 dB. Prior to the formal experiment, participants listened to the white noise at each decibel level through headphones and rated their personal tolerance to each noise pattern on a 5-point Likert scale (0: tolerable; 1: slightly intolerable; 2: moderately intolerable; 3: highly intolerable; 4: extremely intolerable). In the passive stage, the noise rated as “extremely intolerable” (105 dB) by the participants was designated as the high-intensity punishment option, while the noise rated as “slightly intolerable” (70 dB) was chosen as the low-intensity punishment option.

#### 2.1.4. Measures

State Hostility Attribution Bias Scale: The State Hostility Attribution Bias Scale consists of 10 items [35], such as “During the match, the opponent deliberately made things difficult for me.” It uses a 7-point Likert scale, with 1 representing “strongly disagree” and 7 representing “strongly agree,” where higher scores indicate that the participant believed the opponent intentionally harmed them. In this experiment, the Cronbach’s α coefficient for the scale was 0.98.

Positive and Negative Affect Scale (PANAS): The PANAS questionnaire was used to measure the participants’ emotional states [36], including 5 positive adjectives and 5 negative adjectives, namely, happy, joyful, excited, pleased, pleasant, sad, angry, fearful, tense, and upset. It employs a 5-point Likert scale, with 1 indicating “very slight” and 5 indicating “very strong”, where higher scores indicate stronger positive and negative emotions experienced by the participants. In this experiment, the Cronbach’s α coefficient for the positive emotion scale was 0.93, and for negative emotions, the scale was 0.83.

#### 2.1.5. Procedure

The experiment was divided into four stages, including the passive stage, emotional assessment prior to aggression, active stage, and emotional assessment after aggression. Each participant underwent two sessions, experiencing both the low provocation stage and the high provocation stage. The order of the low and high provocation stages was counterbalanced across participants. Each participant engaged in four rounds, with the punisher’s appearance being the same but with different colors in each passive stage, as well as in the active stage, to differentiate between the four opponents.

Stage one was the passive stage, where participants could only choose to flee or endure attacks but could not strike back. Participants were informed in advance that there would be another online matched player serving as the “punisher”, responsible for attacking the participants. The method of attack by the punisher was predetermined based on the manipulation sequence of the provocation types. All of the participants engaged in battles against a computer program. The specific steps were as follows:

Firstly, a fixation point was presented for 1000 ms, followed by the presentation of the selection interface. It displayed the “punisher” choosing between high and low punishment types, with a duration of 1000 ms to 2500 ms.

Secondly, a fixation point was presented for 500–700 ms, followed by the competition interface. In 25 regular trials, a white signal was displayed, and participants needed to quickly press the space bar to react. The rule was that the faster the participant pressed, the better the chance of avoiding punishment. The white signal lasted for 1200 ms and disappeared after the key press. To enhance the authenticity of the battle, if the participant’s key response exceeded 1000 ms, they would definitely receive punishment. In addition, 5 warning trials were randomly inserted into the experiment. After a period of time with the white signal, it would turn into a yellow signal. If the participant pressed the key, they would receive a high punishment, regardless of the punisher’s choice. The initial interval between the white and yellow signals was 300 ms. If the participant did not press the key in the warning trial, the interval between the white and yellow signals would increase by 60 ms in the next trial; if the participant pressed the key in the warning trial, the interval would decrease by 60 ms in the next trial. There was a total of 30 trials in the passive stage.

Thirdly, after a waiting time of 500–1000 ms, the feedback interface was presented for 4000 ms. In regular trials, the punishment type chosen by the opponent and whether the participant accepted the punishment were displayed. If the participant accepted the punishment, they would hear 1000 ms of white noise on this interface. In the warning trials, correctness or incorrectness was displayed, and if incorrect, the participant would also hear 1000 ms of 100 dB white noise. The main experimental procedure followed in the passive phase is shown in Figure 1.

After the passive stage, the State Hostility Attribution Bias Scale was used to measure the perceived hostility of the participants after provocation.

Stage two included an emotional assessment. The PANAS was used to measure the positive and negative emotions of the participants after provocation.

Stage three was the active stage, where the screen randomly matched an opponent as the “recipient”. The participant selected the type of punishment to attack the “recipient”, who could not retaliate. The specific steps were as follows:

Firstly, a fixation point was presented for 1000 ms, followed by the display of the selection interface where the participant chose high or low punishment. The position of high and low punishment was randomly refreshed in each trial.

Secondly, a fixation point was presented for 500–700 ms, followed by the display of the competition interface. Similar to the arousal phase, it included 25 regular trials and 5 warning trials.

Thirdly, after a wait of 500–1000 ms, the feedback interface was presented for 4000 ms. It displayed the punishment type chosen by the participant and whether the punishment was successful. The experimental procedure followed in the aggression phase is shown in Figure 2.

Stage four included an emotional assessment. The PANAS was used to measure the positive emotions and negative emotions of the participants after attacking an innocent person.

At the end of the experiment, the participants were be asked if they were aware that they competed against different opponents in each trial. Data from the participants who did not differentiate between different opponents were be excluded. If a participant answered that they competed against two opponents, they were further asked to guess the identity of the opponent. If the participant answered that the opponent was a computer or AI, the data were be excluded.

### 2.2. Results

#### 2.2.1. Manipulation Check

First, for perceived hostility as the dependent variable, a paired sample *t*-test was conducted for the low and high provocation conditions. The results showed a significant difference in perceived hostility between low and high provocation conditions: *t* = 6.76, *df* = 30, *p* < 0.001, d = 1.21. The perceived hostility in the high provocation condition (*M* = 4.25, *SD* = 2.16) was significantly higher than that in the low provocation condition (*M* = 2.13, *SD* = 1.64).

Next, to verify the relationship between provocation and displaced aggression, the provocation conditions were coded (low provocation = 0; high provocation = 1) for mediation analysis. A mediation effect analysis was conducted with provocation conditions as the independent variable, perceived hostility as the mediator, and displaced aggression as the outcome variable. Using PROCESS in SPSS 23.0, with Bootstrap testing for 5000 samples, after standardizing the scores of perceived hostility and displaced aggression, the results of the mediation analysis showed a significant indirect effect of perceived hostility: *β* = 0.19 and 95% CI = [0.05, 0.35]. The direct effect of provocation was non-significant, *β* = 0.07 and 95% CI = [−0.19, 0.32], indicating that participants’ aggression towards innocents was due to the transfer of hostility, consistent with previous tests of the displaced aggression model [37], as shown in Figure 3.

#### 2.2.2. Effect of Displaced Aggression on Positive Emotion

Descriptive statistics for the mean and standard deviation of positive emotions before and after the aggression are presented in Table 1.

Using positive emotions as the dependent variable, a 2 (provocation) × 2 (measurement phase) repeated measures analysis of variance (ANOVA) was conducted. The results showed that the main effect of provocation was not significant, *F*(1, 30) = 2.18, *p* = 0.15, and $\eta_{p}^{2}$ = 0.07, and the main effect of measurement phase was not significant: *F*(1, 30) = 4.75, *p* = 0.04, and $\eta_{p}^{2}$ = 0.14; however, the interaction between the provocation and measurement phase was significant: *F*(1, 30) = 4.75, *p* = 0.04, and $\eta_{p}^{2}$ = 0.14. Simple main effects analysis revealed that in the low provocation condition, there was no significant difference between pre-test and post-test positive emotions: *F*(1, 30) = 0.01, *p* = 0.92, $\eta_{p}^{2}$ < 0.01; in the high provocation condition, there was a significant difference between pre-test and post-test positive emotions: *F*(1, 30) = 6.45, *p* = 0.02, and $\eta_{p}^{2}$ = 0.18. Pre-test positive emotions were significantly lower than post-test positive emotions. These results indicated that attacking an innocent opponent after high-level provocation enhances positive emotions, as shown in Figure 4.

#### 2.2.3. Effect of Displaced Aggression on Negative Emotion

Descriptive statistics for the mean and standard deviation of negative emotions before and after aggression are presented in Table 2.

Using negative emotions as the dependent variable, a 2 (provocation) × 2 (measurement phase) repeated measures analysis of variance (ANOVA) was conducted. The results showed that the main effect of provocation was significant: *F*(1, 30) = 4.78, *p* = 0.04, and $\eta_{p}^{2}$ = 0.14; the main effect of measurement phase was significant: *F*(1, 30) = 10.33, *p* = 0.003, and $\eta_{p}^{2}$ = 0.26; more importantly, the interaction between the provocation and measurement phase was significant: *F*(1, 30) = 4.84, *p* = 0.04, and $\eta_{p}^{2}$ = 0.14. Simple main effects analysis revealed that in the low provocation condition, there was no significant difference between pre-test and post-test negative emotions: *F*(1, 30) = 0.08, *p* = 0.79, and $\eta_{p}^{2}$ = 0.003; in the high provocation condition, there was a significant difference between pre-test and post-test negative emotions: *F*(1, 30) = 11.43, *p* = 0.002, and $\eta_{p}^{2}$ = 0.28. Pre-test negative emotions were significantly higher than post-test negative emotions. These results indicated that attacking an innocent opponent after high-level provocation moderated negative emotions, as shown in Figure 5.

### 2.3. Discussion

The results of Experiment 1 found that in the high provocation condition, individuals experienced a significant decrease in negative emotions and a significant increase in positive emotions after engaging in displaced aggression, supporting hypothesis H1. Although previous studies have not directly explored the regulatory effects of displaced aggression on emotion in provocative situations, they have investigated the regulatory effects of aggression. Specifically, one study examined the impact of revenge on emotions, dividing participants into two groups: a revenge group read news about Osama bin Laden being arrested and killed, while a control group read neutral news. After that, participants’ positive and negative emotions were then measured, with results showing that the revenge group had significantly higher levels of both positive and negative emotions compared to the control group [38]. In another study, participants were recruited to either perform aggression in a created scenario or act as an audience, after which their positive and negative emotions were measured. The study found that participants who engaged in aggression had significantly higher levels of both positive and negative emotions compared to an audience who did not engage in aggression [39]. Generally, the regulatory effects of aggression on emotion were bitter and sweet. Different from conventional aggression, displaced aggression arises from provocation, but it does not always have a regulatory effect since the target of displaced aggression is often an innocent party. Existing research also indicates that the higher the level of provocation an individual experiences, the higher their levels of displaced aggression [40], and high levels of displaced aggression might help individuals release their own negative emotions to some extent.

However, according to the opponent-process theory, high levels of aggression might not achieve the intended emotion-regulating effects due to the potential triggering of negative emotions, such as guilt. In Experiment 1, participants freely chose the intensity of their aggression, with the overall aggression level falling in the moderate range rather than the high-intensity range. Therefore, Experiment 2 further manipulated the intensity of displaced aggression to examine the effects of high and low levels of displaced aggression on emotions.

## 3. Experiment 2. The Effect of the Intensity of Displaced Aggression on Emotions after Experiencing High Levels of Provocation among Junior High School Students

The results of Experiment 1 suggest that only after high provocation could individuals’ displaced aggression reduce negative emotions and increase positive emotions, and an increase in the proportion of individual aggression could enhance the emotion regulation effect. However, according to opponent-process theory, high-intensity displaced aggression towards innocent parties could generate negative emotions and suppress individuals’ positive emotions. Therefore, Experiment 2 manipulated the intensity of displaced aggression in the highly provocative situation and examined the emotional changes under different levels of aggression to test hypothesis H2.

### 3.1. Methods

#### 3.1.1. Participants

We estimated the required sample size using G*Power 3.1. Experiment 2 utilized a 2 × 2 mixed design, with *f* = 0.25, *α* = 0.05, and a statistical power of 0.8; the estimated sample size was 66 participants. Based on the principle of convenience sampling, seventy-one junior high school students were recruited from a junior high school in Zhuzhou City, Hunan Province, China. According to the experimental requirements, four individuals who did not comply with the requirements were excluded, as well as two individuals who failed to distinguish between the provocateur and the innocent target or doubted the authenticity of the opponent, resulting in a final sample of sixty-five participants included in the analysis. The formal sample consisted of 31 males with an age range of 12–15 years, with a mean age of 12.42 years (*SD* = 0.81), and 34 females with an age range of 12–14 years, with a mean age of 12.41 years (*SD* = 0.7). All of the participants had normal or corrected vision, were not color blind, and had no history of mental illness or surgical trauma. The experiment took place from May to July 2023.

#### 3.1.2. Design

A 2 (intensity: high intensity and low intensity) × 2 (measurement phase: prior to aggression and post-aggression) mixed design was employed, with the intensity as the between-subjects factor and the measurement as the within-subject factor. The low-intensity group consisted of 30 participants, while the high-intensity group consisted of 35 participants. The dependent variables were positive emotions and negative emotions, scored on the positive emotions and negative emotions sub-scales of the PANAS.

#### 3.1.3. Materials and Procedure

We used the same experimental materials and measurements as used in Experiment 1, with the difference being that in Experiment 2, the participants were randomly assigned to the low-intensity aggression group and the high-intensity aggression group. Participants in the low-intensity group were forced to select low punishment, while participants in the high-intensity group were forced to select high punishment.

### 3.2. Results

#### 3.2.1. Manipulation Check about Forced Choice

The proportion of subjects choosing high punishment was used as the dependent variable, and an independent samples *t*-test was conducted between the low-intensity group and the high-intensity group. The results showed a significant main effect of aggressive intensity, *t* = 0.93, *df* = 63, and *p* < 0.01, with the high-intensity group choosing high punishment at proportions (*M* = 0.98 and *SD* = 0.04) significantly higher than the low-intensity group (*M* = 0.04 and *SD* = 0.04). Despite the fact that some participants might have made errors in 1–2 trials by making non-compliant choices, overall, they followed the forced choice principle.

#### 3.2.2. Effect of Displaced Aggressive Intensity on Positive Emotion

Descriptive statistics for the mean and standard deviation of positive emotions before and after the aggression are presented in Table 3.

Using positive emotions as the dependent variable, a 2 (intensity) × 2 (measurement phase) repeated measures analysis of variance (ANOVA) was conducted. The results showed that the main effect of intensity was not significant: *F*(1, 63) < 0.001, *p* = 0.99, and $\eta_{p}^{2}$ < 0.001; the main effect of measurement phase was significant: *F*(1, 63) = 4.73, *p* = 0.033, and $\eta_{p}^{2}$ = 0.07; and the interaction between the intensity and measurement phase was significant: *F*(1, 63) = 7.92, *p* = 0.007, and $\eta_{p}^{2}$ = 0.11. Simple main effects analysis revealed that in the high-intensity group, there was no significant difference between pre-test and post-test positive emotions: *F*(1, 29) = 0.15, *p* = 0.70, and $\eta_{p}^{2}$ = 0.01; in the low-intensity group, there was a significant difference between pre-test and post-test positive emotions, *F*(1, 34) = 17.09, *p* < 0.001, and $\eta_{p}^{2}$ = 0.33, with pre-test positive emotions significantly lower than post-test positive emotions. This indicates that consistently applying low-intensity aggression to innocent individuals can enhance positive emotions, as shown in Figure 6.

#### 3.2.3. Effect of Displaced Aggressive Intensity on Negative Emotion

Descriptive statistics for the mean and standard deviation of negative emotions before and after the aggression are presented in Table 4.

Using negative emotions as the dependent variable, a 2 (intensity) × 2 (measurement phase) repeated measures analysis of variance (ANOVA) was conducted. The results showed that the main effect of intensity was significant: *F*(1, 63) < 0.001, *p* = 0.97, and $\eta_{p}^{2}$ < 0.001; the main effect of measurement phase was significant: *F*(1, 63) = 21.74, *p* < 0.001, and $\eta_{p}^{2}$ = 0.26; however, the interaction between the intensity and measurement phase was not significant, *F*(1, 63) = 0.27, *p* = 0.61, and $\eta_{p}^{2}$ < 0.1, as shown in Figure 7.

#### 3.2.4. Mini Meta-Analysis

We conducted a mini meta-analysis to provide greater transparency in terms of the results and better understand what our findings mean in terms of reliability, replicability, and statistical significance. In total, we calculated two meta-analytical effect sizes on the difference between groups (Cohen’s d) for positive emotions and negative emotions based on the results of Experiment 1 and Experiment 2. A fixed-effects model was used for all meta-analytical effect size calculations [41].

As shown in Table 5, the effects of the displaced aggression on positive emotions and negative emotions in the highly provocative situation were medium and significant. In other words, the results of Experiment 1 and Experiment 2 both indicated that the displaced aggression of individuals in highly provocative situations could reduce the negative emotions and increase the positive emotions.

### 3.3. Discussion

The results of Experiment 2 indicated that, in terms of changes in positive emotions, the low-intensity group showed a significant increase in positive emotions after aggression, while positive emotions of the high-intensity group did not change significantly before and after aggression. In terms of changes in negative emotions, both the low-intensity group and the high-intensity group showed a significant decrease in negative emotions after aggression. The overall results of the experiment suggest that engaging in low-intensity aggression made it easier to regulate emotions, supporting hypothesis H2.

While previous studies have not directly examined the effects of displaced aggression intensity on emotions, existing research on aggression could support the results of this study to some extent. For example, a study introduced the concept of everyday sadism, where participants were asked to choose between aggressive tasks like insect-killing tasks and non-aggressive tasks and reported their emotional states [42]. The results showed that everyday sadism positively predicted the probability of participants choosing aggressive tasks and experiencing pleasure after completing them. This indicated that individuals with higher levels of sadism were more inclined to regulate their emotions through high-intensity aggression, while this might not be the case for individuals with lower levels of sadism.

Additionally, a study on psychopathic killers provided evidence for this, finding that psychopathic killers often exhibited higher levels of gratuitous violence (excessive violence beyond what is necessary to commit murder) and sadistic violence (violence for enjoyment or pleasure) during the committing of crimes [43]. Moreover, individuals with high levels of psychopathy tended to prefer using aggressive forms of humor that direct humorous topics toward harming others to derive pleasure [44]. The study suggested that the phenomenon of experiencing positive emotions through high-level aggression might only occur in special participants with high levels of sadism or psychopathy, while for the general population, the positive emotions of individuals in the high-intensity aggression were not elevated due to societal norms that did not allow high levels of aggression, especially displaced aggression.

## 4. General Discussion

This study investigated the regulatory effects of displaced aggression on emotion among junior high school students in provocative situations through the use of two experiments. The results indicated that displaced aggression could significantly reduce individuals’ negative emotions and increase their positive emotions only in highly provocative situations. Further manipulation of the intensity of aggression in highly provocative situations revealed that only the low-intensity aggression group showed a significant increase in positive emotions after aggression, while both the low-intensity and high-intensity aggression groups exhibited a significant decrease in negative emotions after aggression. This study addressed the previous limitations of indirect evidence regarding the emotional regulation of displaced aggression and effectively expanded on the emotional regulation theory of aggression.

### 4.1. Emotion-Regulating Effects of Displaced Aggression in Provocative Situations

The results of the two experiments both indicated that displaced aggression after provocation could alleviate individuals’ negative emotions and enhance their positive emotions. Specifically, displaced aggression after high-level provocation could reduce negative emotions and increase positive emotions. To some extent, our results support and develop the emotional regulation theory of aggression, suggesting that individuals engage in aggression to regulate emotions, as individuals in a negative emotional state after provocation are motivated to regulate their emotions [45], and aggression is seen as a means to regulate emotions [46], leading individuals to engage in aggression for the purpose of emotional regulation. Previous research has emphasized the regulatory effects of reactive and proactive aggression on emotion, but there has been less focus on the effects of displaced aggression.

On the one hand, displaced aggression could increase positive emotions in junior high school students. Although existing studies have not directly examined the positive emotion-enhancing effects of displaced aggression, research has emphasized that reactive aggression could bring joy through retaliation [47]. Specifically, in this study, participants after provocation could attack the provocateur when they won. Researchers measured the participants’ emotions and recorded the participants’ EEG when they received feedback. The results of the emotional measurements showed that the participants had significantly high positive emotions after retaliation. The EEG results revealed that there was a significant increase in parietal activation at 250–300 ms during the feedback stage in the retaliation condition. This positive wave was related to reward processing, known as reward positivity (RewP), suggesting that the participants’ aggression after provocation was associated with the activation of the reward system. On the other hand, displaced aggression behavior could reduce negative emotions in junior high school students. After provocation, attacking others serves more as catharsis and does not increase negative emotions. A possible explanation was that, after high-level provocation, junior high school students rationalized their displaced aggression towards innocent individuals [48], not seeing their behavior as maladaptive; therefore, they did not experience anxiety, fear, guilt, or other negative emotions due to attacking innocent individuals.

### 4.2. Emotional Regulation Effects of Different Intensities of Displaced Aggression after High-Level Provocation

The results of Experiment 2 indicated that in terms of changes in positive emotions, the low-intensity group showed a significant increase in positive emotions after aggression, while the high-intensity group did not show significant changes in positive emotions before and after aggression. In terms of changes in negative emotions, both the low-intensity group and the high-intensity group showed a significant decrease in negative emotions after aggression, which supported the opponent-process theory to some extent. This theory suggests that initial aggression might elicit negative emotions, prompting individuals to spontaneously generate positive emotions to counteract negative ones. However, the initially generated positive emotions were weak, and with an increasing frequency of aggression, negative emotions decreased while positive emotions increased. In the process of counteracting negative emotions, positive emotions become dominant and become the emotions linked to aggression. However, since the targets of displaced aggression were innocent individuals, it was difficult for individuals to generate positive emotions, and they might have experienced negative emotions, such as guilt. Furthermore, the higher the intensity of aggression, the stronger the negative emotions experienced by individuals.

A previous study found that when sustained high-intensity aggression exceeded the tolerance for retaliation, individuals would not experience positive emotions [49]. Moreover, a study found a reverse U-shaped relationship between aggression intensity and positive emotions, suggesting that moderate-intensity aggression was more effective in regulating emotions [50]. However, another study obtained inconsistent results when using insects as targets of aggression and examined the changes in positive emotions of participants before and after killing insects [51]. The results showed that compared to participants who initially killed one insect, those who initially killed five insects were more likely to kill additional insects later and experienced an increase in positive emotions after aggression. This study indicated that greater aggression intensity was more effective in emotional regulation. However, since the study used insects as the target, it might be difficult for aggressors to empathize with insects. Empathy from aggressors could increase feelings of guilt and reduce positive emotions [52].

## 5. Limitations and Further Direction

This study only used self-report questionnaires to directly measure individuals’ emotional states and did not capture real-time changes in emotions during different time periods. Future studies could consider using physiological arousal indicators of emotions and employ technologies such as affective computing [53] to monitor individuals’ emotions in real time in order to better track the dynamic changes in emotional experiences during aggression and gain more information.

Electrophysiological techniques, such as ERP and fMRI, can also be used to investigate the neurobiological mechanisms underlying the emotional regulation of displaced aggression in provocative situations, revealing the brain activity of individuals regulating emotions through displaced aggression. A previous study on the neural basis of displaced aggression found a significant correlation between the dorsal medial prefrontal cortex and displaced aggression [54]. In early studies, researchers observed the activation of the dorsal medial prefrontal cortex during emoting-regulation processes of participants [55], indicating that the dorsal medial prefrontal cortex might play an important role in regulating emotions through displaced aggression. Future studies could explore the neural mechanisms of emotion regulation through displaced aggression based on the dorsal medial prefrontal cortex, deepening our understanding of how displaced aggression regulates emotions.

Furthermore, although this study found that displaced aggression, especially low-intensity displaced aggression, could regulate individuals’ emotions in highly provocative situations, we did not intervene in the emotion-regulating effects of displaced aggression. Therefore, future research should further explore other alternative intervention measures. While meeting individuals’ emotion-regulating needs, these measures should help individuals acquire positive emotions and reduce negative emotions through some pro-social behaviors advocated by mainstream culture [56].

## 6. Conclusions

This study included two experiments investigating the emotion-regulating effects of displaced aggression among junior high school students after provocation. While supporting and developing the theory of emotion regulation through aggression, the conclusions obtained were as follows:

Firstly, displaced aggression after high-level provocation could alleviate negative emotions and increase positive emotions, but there were no significant changes in negative and positive emotions before and after aggression after low-level provocation.

Secondly, regarding the changes in positive emotions, low-intensity aggression showed a significant increase in positive emotions, while the positive emotions did not show significant changes before and after high-intensity aggression. For changes in negative emotions, both low-intensity and high-intensity displaced aggression significantly decrease negative emotions.

However, considering that displaced aggression violates social norms, efforts should be made to avoid individuals regulating their emotions through displaced aggression, instead guiding them towards using more appropriate methods for emotional regulation in future research and practical applications.

## Figures and Tables

**Figure 1 behavsci-14-00500-f001:**
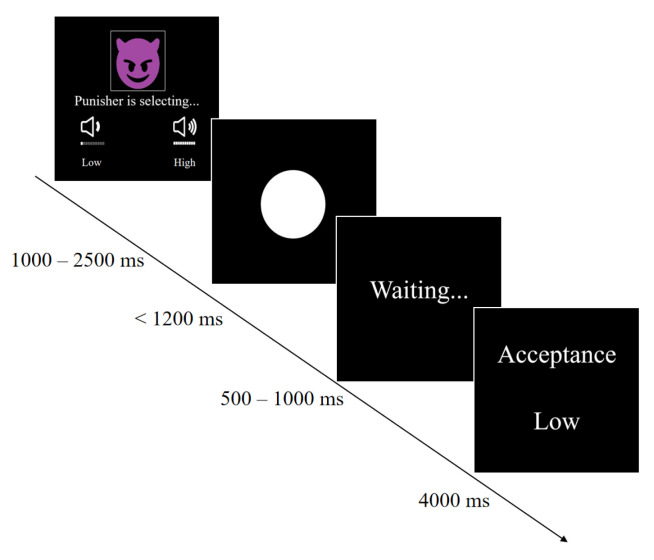
Experimental procedure for the passive stage.

**Figure 2 behavsci-14-00500-f002:**
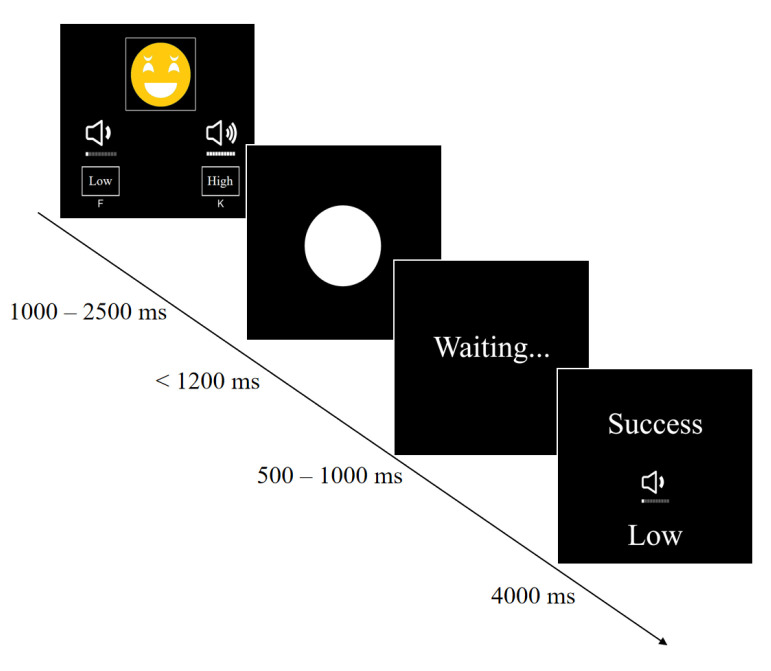
Experimental procedure for the active stage.

**Figure 3 behavsci-14-00500-f003:**
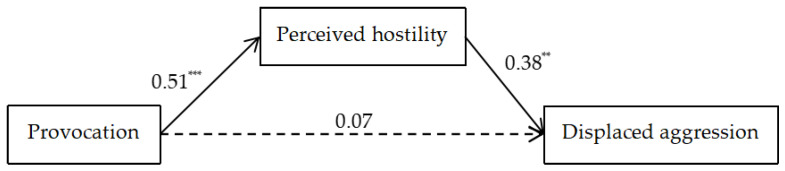
The mediating role of perceived hostility between provocation and displaced aggression. *Note*. ** *p* < 0.01 and *** *p* < 0.001.

**Figure 4 behavsci-14-00500-f004:**
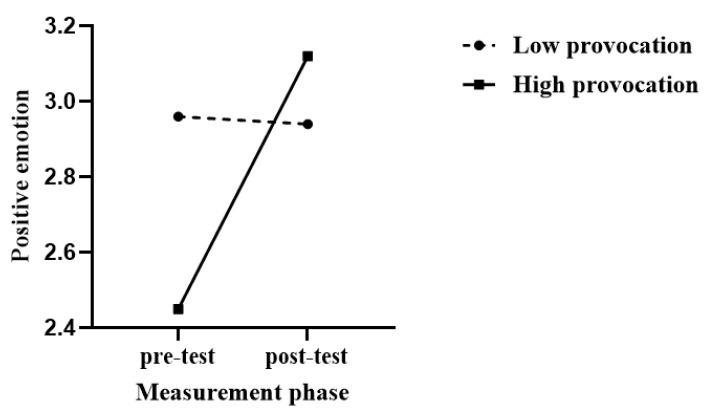
Effects of displaced aggression on positive emotion in high and low provocation.

**Figure 5 behavsci-14-00500-f005:**
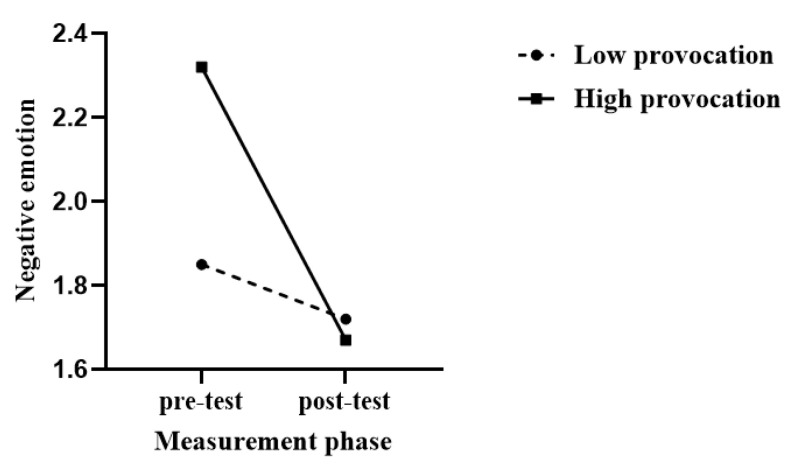
Effects of displaced aggression on negative emotion in high and low provocation.

**Figure 6 behavsci-14-00500-f006:**
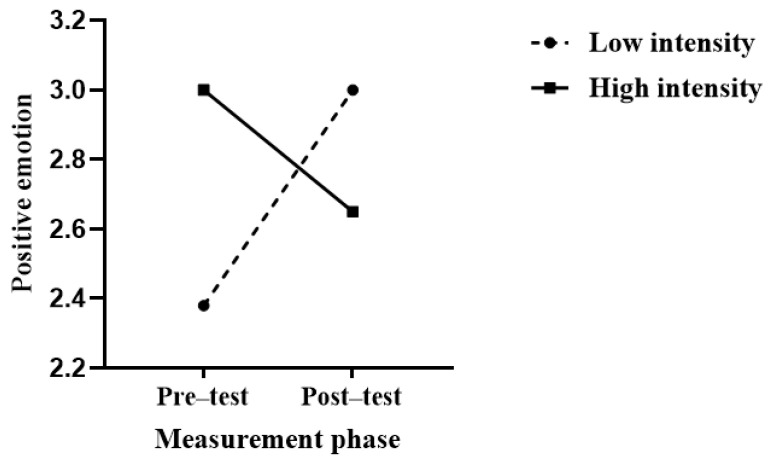
Effects of displaced aggression on positive emotion in high and low intensity condition.

**Figure 7 behavsci-14-00500-f007:**
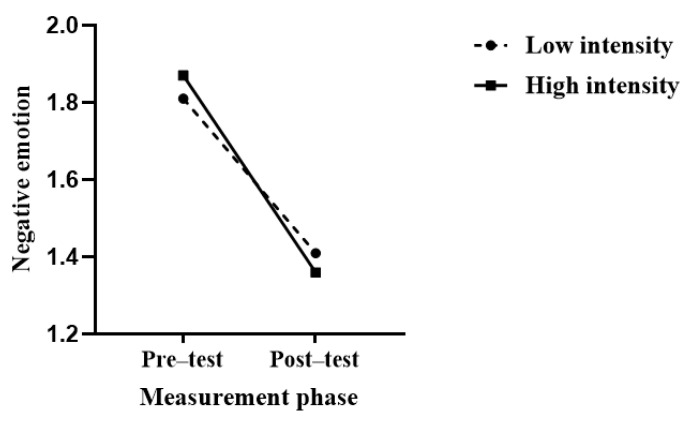
Effects of displaced aggression on negative emotion in high and low intensity condition.

**Table 1 behavsci-14-00500-t001:** The mean (standard deviation) of positive emotion in high and low provocation condition.

		Measurement Phase	
		Prior to Aggression	Post Aggression
Provocation	Low	2.96 (1.25)	2.94 (1.28)
	High	2.45 (1.22)	3.12 (1.31)

**Table 2 behavsci-14-00500-t002:** The mean (standard deviation) of negative emotion in each condition in high and low provocation condition.

		Measurement Phase	
		Prior to Aggression	Post Aggression
Provocation	Low	1.85 (0.58)	1.72 (0.89)
	High	2.32 (1.04)	1.67 (0.95)

**Table 3 behavsci-14-00500-t003:** The mean (standard deviation) of positive emotion in high and low intensity condition.

		Measurement Phase	
		Prior to Aggression	Post Aggression
Intensity	Low	2.38 (1.24)	3.00 (1.41)
	High	2.73 (1.2)	2.65 (1.37)

**Table 4 behavsci-14-00500-t004:** The mean (standard deviation) of positive emotion in each condition in high and low intensity condition.

		Measurement Phase	
		Prior to Aggression	Post Aggression
Intensity	Low	1.81 (0.15)	1.41 (0.09)
	High	1.87 (0.15)	1.36 (0.06)

**Table 5 behavsci-14-00500-t005:** Estimates of overall effects of the displaced aggression on positive emotion and negative emotion in highly provocative situations based on mini meta-analyses of Experiment 1 and Experiment 2.

	Standardized Difference in Means (Cohen’s *d*)	95% Confidence Interval		*p*
Emotion		Lower limit	Upper limit	
Positive Emotion	0.34	0.13	0.54	0.001
Negative Emotion	−0.59	−0.81	−0.37	<0.001

## Data Availability

The research data will be available upon request to the corresponding author.

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
