# Peer review of "Emotional Regulation of Displaced Aggression in Provocative Situations among Junior High School Students"

_behavsci, 2024, doi:10.3390/bs14060500_

Round 1

Reviewer 1 Report

Comments and Suggestions for Authors

The topic is interesting, but in the reviewer's opinion, it needs at least some clarifications and additions.

1.I understand the requirements of abstracts, but isn't at least one sentence about the research needed when the conclusions are so general.

2.A considerable number of footnotes contain rather distant literature. See, for example, footnotes 1, 2, 3, 30, 45 and others. One would expect an explanation as to the distant literature in time.

3.In many places, there is a disparate notation of footnotes. Sometimes it is justified, sometimes it causes difficulties in reading. See, for example, line 50, 566. The entire text should be reviewed in this regard.

4 The reading of the text is made more difficult if we read it together with the footnotes. In line 65, we read about recent research and have a footnote to a journal from 2019. In line 73 we have a footnote to a text from 2017, and in footnote 21 we have a footnote to 2008. How justified is it to cite such distant studies while maintaining continuity of thought.

5. In footnote 17 the author's text is from 2017. Then we read about previous studies and we have a footnote from 2023. The article needs to be reviewed in this regard.

6. In the research methodology, we read about a certain school in a certain province (line 85, 421).However, shouldn't more specific data be given, the size of the city in some range. What determined the choice of such a city and what was the key to the selection of the respondents.

7. It should be explained why 36 students participated in the first survey and 71 in the second.

8. It is not stated in the text when the survey took place. If it was the time of the Covid 19 pandemic, how valid are the survey results?

9. Are the stated limitations of the research and the prospect of future research sufficient (lines 577 to 581). 

Comments on the Quality of English Language

Pay attention to the correctness of the recording of the times 

Author Response

Dear reviewer,

We sincerely appreciate your valuable comments and suggestions on our manuscript “Emotional regulation of displaced aggression in provocative situations among junior high school students ”(behavsci-2991016). Based on your comments, we have made revision to our manuscript as described below.

Point 1: Abstract: I understand the requirements of abstracts, but isn't at least one sentence about the research needed when the conclusions are so general. 

Response 1: Thanks for your suggestion, we have expanded and reorganized the content of the conclusions, as shown in the blue font in the abstract.

Point 2: References: A considerable number of footnotes contain rather distant literature. See, for example, footnotes 1, 2, 3, 30, 45 and others. One would expect an explanation as to the distant literature in time. In many places, there is a disparate notation of footnotes. Sometimes it is justified, sometimes it causes difficulties in reading. See, for example, line 50, 566. The entire text should be reviewed in this regard. The reading of the text is made more difficult if we read it together with the footnotes. In line 65, we read about recent research and have a footnote to a journal from 2019. In line 73 we have a footnote to a text from 2017, and in footnote 21 we have a footnote to 2008. How justified is it to cite such distant studies while maintaining continuity of thought. In footnote 17 the author's text is from 2017. Then we read about previous studies and we have a footnote from 2023. The article needs to be reviewed in this regard.

Response 2: Thank you for your comments. We have checked all references in the manuscript, restructured the logical flow of the introduction and discussion sections, and replaced some old references, particularly empirical research studies, with more recent research, as indicated by the blue font in the main text and reference section.

Point 3: Methods: In the research methodology, we read about a certain school in a certain province (line 85, 421).However, shouldn't more specific data be given, the size of the city in some range. What determined the choice of such a city and what was the key to the selection of the respondents.

Response 3: Thank you for your suggestion on increasing the subject information. Based on the principle of convenience sampling, 36 junior high school students were recruited from a junior high school in Zhuzhou City, Hunan Province, China. Zhuzhou City is an ordinary city with a population of around 3.85 million, which GDP arrive at medium level among all cities of China.  We selected an average junior high school in Zhuzhou City, where the students' academic performance is at a medium level compared to the rest of the city.

Point 4: Methods: It should be explained why 36 students participated in the first survey and 71 in the second.

Response 4: Thank you for your comments. Experiment 1 employed a within-subject design with a 2(provocation: high provocation, low provocation) × 2 (measurement phase: pre-test, post-test), and sample size calculation for a 2 × 2 two-factor within-subject design was conducted using G*Power. With f = 0.25, α = 0.05, and a statistical power of 0.8, the estimated sample size was 24. Experiment 1 included 31 participants for formal analysis, meeting the sample size requirements. Experiment 2 utilized a mixed design with a 2(intensity: high intensity, low intensity) × 2 (measurement phase: pre-test, post-test), and sample size calculation for a 2 × 2 mixed design was performed using G*Power. With f = 0.25, α = 0.05, and a statistical power of 0.8, the estimated sample size was 66. Experiment 2 included 65 participants for formal analysis, which basically met the sample size requirements.

Point 5: Methods: It is not stated in the text when the survey took place. If it was the time of the Covid 19 pandemic, how valid are the survey results?

Response 5: Thank you for your comments. Our study was not conducted during the COVID-19 pandemic. The study period was from April to July, 2023, with Experiment 1 conducted from April to May, 2023, and Experiment 2 conducted from May to July, 2023.

Point 6: Limitation: Are the stated limitations of the research and the prospect of future research sufficient (lines 577 to 581).

Response 6: Thank you for your suggestions. We have expanded the limitations and prospects of the study, as detailed in the blue font in Section 5 on research limitations and prospects.

Point 7: Quality of English Language: Pay attention to the correctness of the recording of the times.

Response 7: Thank you for your suggestions. We have reviewed and revised the English expressions through MDPI’s english editing service, as indicated by the blue font in the document.

We sincerely appreciate your valuable comments and suggestions. Should our revisions be inadequate or incorrect, we kindly ask for the opportunity to further revise the manuscript please. Your feedback is invaluable, and we are grateful for your assistance and support throughout this process. Thank you once again for your guidance and support.

Reviewer 2 Report

Comments and Suggestions for Authors

In statistical analysis and research, the representativeness of the sample is a crucial aspect. The present study is carried out through the quantitative paradigm using a sample of 36/71 subjects from a very specific situation. Therefore, the sample is not representative enough to be considered reliable. The generally accepted rule is 95% confidence and 5% sampling error. It is necessary to determine the population of the area studied and to ascertain whether this rule holds true.

The inadequacy of the sample has an impact on two fundamental issues:

1. Reliability of the results: Representative samples collect data that can be considered representative of the wider population. This is essential for market research and statistical analysis.

2. Avoiding bias: Without representativeness, data can be useless or biased. If we do not include the relevant characteristics in the sample, the results may be misleading.

In the event that the research is intended to be a case study, the most appropriate methodology would be qualitative, given that this paradigm already contemplates the case study and admits samples of this size.

 Conversely, it is not logical to present an article in a segregated manner. If two experiments with a common theoretical framework are conducted and are to be presented in the same article, they should be analysed together, with a joint methodological framework. This method of presenting the research allows for the identification of easily solvable flaws, such as the numbering of the sections. Furthermore, it requires the reader to expend additional effort in reading, which adds unnecessary complexity to the manuscript.

Author Response

Dear reviewer,

We sincerely appreciate your valuable comments and suggestions on our manuscript “Emotional regulation of displaced aggression in provocative situations among junior high school students ”(behavsci-2991016). Based on your comments, we have made revision to our manuscript as described below.

Point 1: In statistical analysis and research, the representativeness of the sample is a crucial aspect. The present study is carried out through the quantitative paradigm using a sample of 36/71 subjects from a very specific situation. Therefore, the sample is not representative enough to be considered reliable. The generally accepted rule is 95% confidence and 5% sampling error. It is necessary to determine the population of the area studied and to ascertain whether this rule holds true.

The inadequacy of the sample has an impact on two fundamental issues:

  1. Reliability of the results: Representative samples collect data that can be considered representative of the wider population. This is essential for market research and statistical analysis.

  1. Avoiding bias: Without representativeness, data can be useless or biased. If we do not include the relevant characteristics in the sample, the results may be misleading.

In the event that the research is intended to be a case study, the most appropriate methodology would be qualitative, given that this paradigm already contemplates the case study and admits samples of this size.

Response 1: Thank you for your comments. In this study, we recruited participants from a junior high school using convenience sampling. Since we recruited junior high school students from normal junior high schools, we assumed that population of current study had general characteristics. To ensure the unbiasedness of the experimental sample in this study, we conducted the following tests.

Firstly, regarding the representativeness of the sample in the current study, we previously conducted a questionnaire on displaced aggression tendencies to junior high school students in two junior high schools in the provinces of Guangxi and Hunan, China [1]. The sample size was N = 2095, with 1082 males (51.65%) and 1013 females (48.35%), aged 11-15 years, with a mean age of Mmale = 12.78 (SD = 0.63) and Mfemale = 12.61 (SD = 0.60). We conducted goodness-of-fit tests on the samples of Experiment 1 and Experiment 2, using the population's quartile deviation as the threshold to divide the samples. In terms of displaced aggression tendencies, the goodness-of-fit test for Experiment 1 yielded χ2 = 4.23, df = 3, p = 0.24, indicating that the Experiment 1 sample matched the population distribution. Similarly, the goodness-of-fit test for Experiment 2 resulted in χ2 = 4.72, df = 3, p = 0.19, indicating that the Experiment 2 sample matched the population distribution. Furthermore, we conducted a homogeneity test on the high-punishment group and low-punishment group in Experiment 2. Using displaced aggression tendency scores as the dependent variable, an independent samples t-test was performed between the two groups, t = 0.93, df = 63, p = 0.36, indicating homogeneity in displaced aggression tendencies between the two groups of participants.

Secondly, we balanced the gender and age of participants in Experiment 1 and Experiment 2. In Experiment 1, there were 17 males (54.84%) among 31 participants. Additionally, an independent samples t-test on the ages of males and females in Experiment 1 revealed no significant difference, t = 0.26, df = 29, p = 0.079. In Experiment 2, there were 31 males (47.69%) among 65 participants. Among them, the low-punishment group included 15 males (42.86%) and the high-punishment group included 16 males (53.33%). Furthermore, we classified and described the age of participants in each group based on gender and conducted descriptive statistics, as shown in Table 1.

Table 1. The mean (standard deviation) of age characteristics of male and female junior high school students in the high-punishment group and low-punishment group

Gender

Male

Female

Intensity

Low

12.4 (0.83)

12.3 (0.57)

High

12.44 (0.81)

12.57 (0.85)

Using age as the dependent variable, a 2 × 2 factorial analysis of variance revealed that the main effect of gender was not significant, F(1,61) = 0.01, p = 0.93; the main effect of group was not significant, F(1,61) = 0.66, p = 0.42; and the interaction effect between gender and group was not significant, F(1,61) = 0.38, p = 0.54. This indicated that we matched ages of the participants effectively.

To obtain more reliable statistical results, we estimated the required sample size for Experiment 1 and Experiment 2 using G*Power. For Experiment 1, employing a 2 × 2 within-subject design, with f = 0.25, α = 0.05, and a statistical power of 0.8, the estimated sample size was 24. Experiment 1 included a total of 31 participants for formal analysis, meeting the sample size requirement. For Experiment 2, utilizing a 2 × 2 mixed design, with f = 0.25, α = 0.05, and a statistical power of 0.8, the estimated sample size was 66 participants. Experiment 2 included 65 participants for formal analysis, which essentially met the sample size requirement.

  1. Sun, S, N.; Cheng, G, C.; Bai, X, J.; Feng, M, M.; Lin, S. The relationship between harsh parenting and junior high school students’ displaced aggression: A moderated mediation model. Curr Psychol. Under review.

Point 2: Conversely, it is not logical to present an article in a segregated manner. If two experiments with a common theoretical framework are conducted and are to be presented in the same article, they should be analysed together, with a joint methodological framework. This method of presenting the research allows for the identification of easily solvable flaws, such as the numbering of the sections. Furthermore, it requires the reader to expend additional effort in reading, which adds unnecessary complexity to the manuscript.

Response 2: Thank you for your comments. Regarding the common methodological framework, we have supplemented the role of emotional regulation theory in our study in the Introduction and Discussion sections, as indicated by the blue font in these sections. Additionally, concerning the common methodological framework, we added a mini meta-analysis of the two experiments in the Results section of Experiment 2, as shown in blue font in the Experiment 2 Results section.

We sincerely appreciate your valuable comments and suggestions. Should our revisions be inadequate or incorrect, we kindly ask for the opportunity to further revise the manuscript please. Your feedback is invaluable, and we are grateful for your assistance and support throughout this process. Thank you once again for your guidance and support.

Reviewer 3 Report

Comments and Suggestions for Authors

I thank the editor for the opportunity to review this manuscript. This paper aims, through two separate experiments, to determine the effect of emotion regulation of displaced aggression among high school students. It may be a work of relevance to the scientific community, but in its current version it has some limitations.

1. The format of references needs to be revised. It is not correct, for example: line 50, line 127…

2. This may depend on the preferences of the individual researcher. It is true that in the social sciences, the scientific community tends to give excessive explanations of the studies it carries out and has to make an extra effort to justify the adequacy and results of its studies in comparison to other sciences. I mean, there is currently an introduction of around 2000 words, if I am not mistaken. This makes the introduction very difficult to read and understand, and it loses the essence of "justification" that it should have in order to be part of a scientific study. I would recommend the authors to rephrase the introduction and shorten it as much as possible.

3. Presenting two separate studies does not mean that there is no common design and method. I suggest creating a method that unifies both studies.

4. Adding a figure to explain mediation in lines 332-342 would be interesting.

5. Science must be able to offer its knowledge in conditions of transfer to society. Could you add some idea of what this study brings to society in the “Conclusions” section?

Comments on the Quality of English Language

In relation to English, it is still a scientific English. I would recommend a further revision to simplify the wording of the manuscript.

Author Response

Dear reviewer,

We sincerely appreciate your valuable comments and suggestions on our manuscript “Emotional regulation of displaced aggression in provocative situations among junior high school students ”(behavsci-2991016). Based on your comments, we have made revision to our manuscript as described below.

Point 1: References: The format of references needs to be revised. It is not correct, for example: line 50, line 127…

Response 1: Thank you for your comments. We have modified all the references, as shown in the blue part of the paper.

Point 2: Introduction: This may depend on the preferences of the individual researcher. It is true that in the social sciences, the scientific community tends to give excessive explanations of the studies it carries out and has to make an extra effort to justify the adequacy and results of its studies in comparison to other sciences. I mean, there is currently an introduction of around 2000 words, if I am not mistaken. This makes the introduction very difficult to read and understand, and it loses the essence of "justification" that it should have in order to be part of a scientific study. I would recommend the authors to rephrase the introduction and shorten it as much as possible.

Response 2: Thank you for your advice. We have made modifications and simplifications to the wording in the Introduction section, as indicated by the blue font in that section.

Point 3: Methods: Presenting two separate studies does not mean that there is no common design and method. I suggest creating a method that unifies both studies.

Response 3: Thank you for your suggestion. We have added a mini meta-analysis of the two experiments in the results section of experiment 2, as indicated by the blue font in the section on 3.3.2 Mini Meta-Analysis.

Point 4: Results: Adding a figure to explain mediation in lines 332-342 would be interesting.

Response 4: Thank you for your advice. Your suggestions have been very helpful in our modifications. We have added a diagram illustrating the mediating effect, as shown in Figure 3.

Point 5: Conclusions: Science must be able to offer its knowledge in conditions of transfer to society. Could you add some idea of what this study brings to society in the “Conclusions” section?

Response 5: Thank you for your suggestion. We have expanded the conclusion section as per your request, as indicated by the blue font in the conclusion section.

Point 6: Quality of English Language: In relation to English, it is still a scientific English. I would recommend a further revision to simplify the wording of the manuscript.

Response 6: Thank you for your suggestions. We have reviewed and revised the English expressions through MDPI’s english editing service, as indicated by the blue font in the document.

We sincerely appreciate your valuable comments and suggestions. Should our revisions be inadequate or incorrect, we kindly ask for the opportunity to further revise the manuscript please. Your feedback is invaluable, and we are grateful for your assistance and support throughout this process. Thank you once again for your guidance and support.

Round 2

Reviewer 1 Report

Comments and Suggestions for Authors

I accept after corrections and clarifications.

Reviewer 2 Report

Comments and Suggestions for Authors

The authors have adequately justified the issues raised in the previous review.